# Secondary Metabolism Gene Clusters Exhibit Increasingly Dynamic and Differential Expression during Asexual Growth, Conidiation, and Sexual Development in *Neurospora crassa*

Zheng Wang,[a] Francesc Lopez-Giraldez,[b] Jason Slot,[c] Oded Yarden,[d] Frances Trail,[e] Jeffrey P. Townsend[a,f,g]

[a]Department of Biostatistics, Yale University, New Haven, Connecticut, USA

[b]Yale Center for Genome Analysis (YCGA), Department of Genetics, Yale University, New Haven, Connecticut, USA

[c]Department of Plant Pathology, The Ohio State University, Columbus, Ohio, USA

[d]Department of Plant Pathology and Microbiology, The Robert H. Smith Faculty of Agriculture, Food, and Environment, The Hebrew University of Jerusalem, Rehovot, Israel

[e]Department of Plant, Soil, and Microbial Sciences, Michigan State University, East Lansing, Michigan, USA

[f]Department of Ecology and Evolutionary Biology, Yale University, New Haven, Connecticut, USA

[g]Program in Computational Biology and Bioinformatics, Yale University, New Haven, Connecticut, USA

**ABSTRACT** Secondary metabolite clusters (SMCs) encode the machinery for fungal toxin production. However, understanding their function and analyzing their products requires investigation of the developmental and environmental conditions in which they are expressed. Gene expression is often restricted to specific and unexamined stages of the life cycle. Therefore, we applied comparative genomics analyses to identify SMCs in *Neurospora crassa* and analyzed extensive transcriptomic data spanning nine independent experiments from diverse developmental and environmental conditions to reveal their life cycle-specific gene expression patterns. We reported 20 SMCs comprising 177 genes—a manageable set for investigation of the roles of SMCs across the life cycle of the fungal model *N. crassa*—as well as gene sets coordinately expressed in 18 predicted SMCs during asexual and sexual growth under three nutritional and two temperature conditions. Divergent activity of SMCs between asexual and sexual development was reported. Of 126 SMC genes that we examined for knockout phenotypes, *al-2* and *al-3* exhibited phenotypes in asexual growth and conidiation, whereas *os-5*, *poi-2*, and *pmd-1* exhibited phenotypes in sexual development. SMCs with annotated function in mating and crossing were actively regulated during the switch between asexual and sexual growth. Our discoveries call for attention to roles that SMCs may play in the regulatory switches controlling mode of development, as well as the ecological associations of those developmental stages that may influence expression of SMCs.

**IMPORTANCE** Secondary metabolites (SMs) are low-molecular-weight compounds that often mediate interactions between fungi and their environments. Fungi enriched with SMs are of significant research interest to agriculture and medicine, especially from the aspects of pathogen ecology and environmental epidemiology. However, SM clusters (SMCs) that have been predicted by comparative genomics alone have typically been poorly defined and insufficiently functionally annotated. Therefore, we have investigated coordinate expression in SMCs in the model system *N. crassa*, and our results suggest that SMCs respond to environmental signals and to stress that are associated with development. This study examined SMC regulation at the level of RNA to integrate observations and knowledge of these genes in various growth and development conditions, supporting combining comparative genomics and inclusive transcriptomics to improve computational annotation of SMCs. Our findings

Address correspondence to Jeffrey P. Townsend, jeffrey.townsend@yale.edu.

The authors declare no conflict of interest.

call for detailed study of the function of SMCs during the asexual-sexual switch, a key, often-overlooked developmental stage.

**KEYWORDS** *Neurospora crassa*, asexual development, environmental microbiology, filamentous fungi, gene cluster, secondary metabolism, sexual development, transcriptomics

The kingdom Fungi exhibits some of the largest species and ecological diversity on the planet (1–3). Many fungi and fungal products are socially and economically important: the presence of fungi and consequent by-products can lead to beneficial or harmful effects on the environment (4–6). Among the fungal products responsible for these effects, secondary metabolites (SMs) are abundant and diverse in fungi (5, 7, 8). Many of these low-molecular-weight compounds can functionally affect their environment, hosts, and human life and are critical for the survival of their producers (9–11). Our knowledge about fungal SMs derives mostly from studies on a few fungal species of significance to agriculture and medicine—studies that have been focused on a few specific chemical products under optimized conditions (e.g., 12–17). Burgeoning fungal genome data have enabled the rapid identification of SM genes and gene clusters and made possible their annotation with diverse functions in fungal biology and ecology (5, 7, 18). However, many genes in SMCs are silent or expressed at very low levels in standard laboratory conditions (7), and the identification and isolation of secondary metabolites requires knowledge of their robust expression (19). Therefore, systematic investigation of SM expression dynamics across all key stages of the life cycle in diverse environmental conditions is essential. Such investigations in a fungal model can provide insights into functional and regulatory roles of SMCs across the fungal life cycle.

Most fungal SMs can be classified into four chemical classes or combinations within them—polyketides, terpenoid, nonribosomal peptides, and shikimic acid compounds (5). Genes that encode signature enzymes for these products are known as polyketide synthases (PKS), nonribosomal peptide synthases (NRPS), tryptophan synthetases (TS), and dimethylallyl tryptophan synthetases (DMATS). Within the fungal kingdom, filamentous ascomycete genomes tend to harbor more genes for secondary metabolism than genomes in other fungal lineages (20), whereas some fungal groups are not known to possess any secondary metabolites (21). Genomes of ascomycetes that have been made available by NCBI and the JGI 1000 Fungal Genome project encode different numbers of enzymes for synthesis of PKS, NRPS, TS, and DMATS (9). Within Ascomycetes, Sordariomycetes host an impressive ecological diversity and an exceptional number of species that are under investigation for genes in SM production (5, 22–25). In most cases these SM genes are annotated on chromosomes as "secondary metabolism" gene clusters (SMCs, sometimes referred to as biosynthetic gene clusters), and they are typically responsible for the anabolic chemistry of secondary metabolites as well as their regulation and transport (26, 27). SMCs are usually identified with bioinformatic approaches based on gene sequence similarity and conservation of gene colocation (synteny), which is aided by fungal diversity. However, extensive genetic and molecular biological knowledge of model fungi facilitates the understanding of gene activities and their regulation, including epigenetic regulation of gene expression, and thereby facilitates identification and functional annotation of SMCs. This regulatory information can be applied toward understudied fungi that represent a huge diversity in ecology compared with more limited numbers of well-studied reference genomes (10, 28). Therefore, studies on genomic regulation of SMCs and the mycotoxins they produce can be facilitated by the availability of genomic and transcriptomic data from models as well as nonmodel fungi.

Environmental factors are known to regulate production of SMs in diverse fungal species, especially via transcriptional regulation by transcription factors or epigenetic mechanisms (7, 10, 29–32). For example, environmental changes can induce fungal SM toxins, pigments, and other chemicals in association with fungal stress responses to

drought, reactive oxygen species, temperature, and UV light intensity (9, 33–35). Fungal toxin production is known to respond to the presence of host elements, including the presence of other fungi, and in the laboratory it is often manipulated by culture on specialized media (36–39). Genome-wide regulation of secondary metabolites during the production of conidia (mitotic spores) has only recently been investigated in fungal genomes enriched with SMCs, including *Aspergillus* spp. (40–43). In addition, nutrient levels are essential signals to induce switches in reproductive mode or pathogen virulence of many fungi. Some studies suggested nitrogen limitation and amino acid levels affect SM biosynthesis, especially synthesis by NRPSs (44–47). However, environmental factors, including nutrients, are also key regulators for development and reproduction in many fungi. Genes regulating sporulation are known to affect some SMCs (48–51). Therefore, investigation of the expression of SMCs across fungal development in additional environmental contexts provides an opportunity to enhance our understanding of their regulation and function.

Environmental impacts on fungal growth and development have been a major focus of research on the model species *Neurospora crassa* (52–57). Long a genetic model system, *N. crassa* was the first filamentous fungus to have its genome publicly sequenced (55, 58). *Neurospora* species are known to produce toxins, such as neurosporin A and the salicylaldehyde sordarial, as a defense mechanism (59, 60). As in many filamentous fungi, the life cycles of *Neurospora* species can be divided into three well-studied developmental phases: asexual growth and reproduction, the asexual-sexual transition, and sexual development and reproduction (50, 56, 57, 61–73). The ease with which it can be manipulated under laboratory conditions, its distinctive morphology for establishing developmental check-points, its well-annotated genome, the modest complement of SMCs in its genome, and the extensive corpus of gene expression data previously collected make *Neurospora crassa* a convenient model for tracking SMC activity throughout the life cycle under different laboratory conditions.

In this study, we performed an *in silico* genome-wide search for SMCs in the *N. crassa* genome, extending the known list of SMCs and their core genes in this species. To investigate divergent regulation of SMC activities across developmental stages of the *N. crassa* life cycle, we revisited previous RNAseq data on key stages of asexual growth and reproduction, including conidial germination and conidiation, as well as sexual development and reproduction. To explore how SMC activities diverge in response to additional developmental and ecological factors, we analyzed previous data regarding the impact of light on *N. crassa* transcriptomics as well as new transcriptomic data from the *N. crassa* conidial germination process at high temperature (37°C) and on the commonly used (but nonstandard for *N. crassa*) fungal medium potato dextrose agar (PDA). These analyses provided insight into how expression of SMCs and production of mycotoxin are associated with growth, developmental status, and environmental condition. The congruence of RNA and protein expression levels has been controversial (74). Therefore, rather than assess absolute expression levels, we evaluated the coordinated regulation of RNA expression among all genes in each SMC across a suite of independent experiments in distinct environmental and developmental conditions. Significant coordination in expression for multiple genes in an SMC indicates these genes are under unified regulation under the condition being sampled. The term "coordinately regulated gene sets" (CRGS) has been applied to describe clustered sets of genes within SMCs that exhibit congruent RNA expression (75). We define CRGSs as SMCs within which the majority of genes (>50%) exhibit congruent expression. Identification of a CRGS is indicative of SMCs that are active under specific environmental/developmental conditions.

## RESULTS AND DISCUSSION

We investigated three aspects of the systemic regulation of all SMCs in a model fungus (Table 1): the number of SMCs and their contents based on the comparative genomics, the activities of SMCs based on high-quality transcriptomics under different conditions

**TABLE 1** Summary of the data, analyses, and results in this study[a,b]

| Data | Methods | Detailed results |
|---|---|---|
| Genomics[a] | | |
| *N. crassa*[b] | AntiSMASH (v5) and JGI database | 20 SMCs with 177 genes were identified in these |
| *N. discreta* | | genomes (Table 2) |
| *N. tetrasperma* | | |
| Transcriptomics | | |
| *N. crassa* asexual growth (BM 25°C) | LOX normalization | Fold-changes and mapped counts (Tables S1 and S2) |
| *N. crassa* asexual growth (BM 37°C) | Pearson correlation (R) | Correlations among SMCs (Fig. 1 and 6, Table S3) |
| *N. crassa* asexual growth (MM 25°C) | LPWC correlation | CRGs (Fig. S1, Tables S4 and S5) |
| *N. crassa* asexual growth (PDA 37°C) | | |
| *N. crassa* growth respond to light[c] | | |
| *N. crassa* conidiation (mix)[d] | | |
| *N. crassa* sexual reproduction (SMC) | | |
| Knockouts | | |
| *N. crassa* knockout mutants on BM | Phenotyping during asexual and | 126 SMC genes were examined for KO phenotypes |
| and SCM (25°C) | sexual growth | (Fig. S1, Fig. 7) |

[a]Genome annotations were derived from the JGI MycoCosm database (https://mycocosm.jgi.doe.gov/mycocosm/home). For execution of the AntiSMASH fungal version, the latest annotation of the *N. crassa* genome was supplied, and detection stringency was set to be "relaxed," with extra features applied, including known cluster BLAST, subcluster BLAST, cluster Pfam analysis, active site finder, and RREfinder, as well as cluster border prediction based on transcription binding sites.

[b]The *N. crassa mat A* wild-type Oak Ridge (OR) strain, obtained from the Fungal Genetic Stock Center (FGSC2489, which other than at the mating-type locus has the same genetic background as the *mat a* wild-type strain available as FGSC4200) was studied in the experiments under constant light in the Townsend laboratory.

[c]RNAseq data of *N. crassa* from liquid culture of hyphae of wild-type strains (FGSC2489 and -4200) in response to light exposure (from Chen et al. [94]). Data points include 0 (dark), 15, 60, 120, and 240 min of exposure to light after a 24-h dark treatment.

[d]RNAseq data of *N. crassa* conidiation from Sun et al. (84) were reanalyzed, including three stages of *N. crassa* conidiation (wild-type *mat a* strain FGSC4200) that were sampled from culture on Vogel's medium under constant light. The sexual development stage sampled in Sun et al. (84) was excluded from transcriptomic analysis in this study to avoid an inappropriate mixture of heterogenous stages of the life cycle within our analysis.

across the fungal life cycle, and the possible roles of SMCs based on knockout phenotypes. Data analysis was performed using the fungal version of antiSMASH v5.0 (76), LOX v1.6 for comparative gene expression (77), an R package (Corr) for Pearson correlation, and lag-penalized weighted correlations (LPWC; 78). All of these programs provide a strong basis for statistical inferences. With one exception (a reanalysis of data on conidiation from another research group), data normalization and analysis were restricted within the experiment of the same computational settings and experimental platforms.

**Secondary metabolism gene clusters (SMCs) were recognized.** A total of 20 SMCs with 177 genes were recognized and analyzed (Table 2). Out of the 20 SMCs, 19 SMCs were recognized within the *N. crassa* genome by antiSMASH (76) and 1, composed of NCU07307 and -07308, was previously identified in the JGI MycoCosm fungal genome database (18). Among the 19 SMCs predicted by antiSMASH, 16 featured core genes that had also been previously identified using a different algorithm utilized in the JGI MycoCosm database. Orthologs of these core genes in all 19 SMCs were identified in *N. crassa*, *Neurospora tetrasperma*, and *Neurospora discreta*. antiSMASH predicted five additional SMCs that were previously unreported in the JGI MycoCosm database.

The predicted SMCs are located on six of the seven *N. crassa* chromosomes. Only chromosome III (supercontig NC_026503) exhibited no SMCs. Among the 174 SMC genes, 118 have not yet been annotated for their functions. Among the genes in SMCs predicted with antiSMASH, seven genes were only found in *Neurospora* genomes, including NCU00583 (SMC no. 1), -04804 (SMC no. 13), -04867 (SMC no. 7), -04998 (SMC no. 16), -05759 (SMC no. 20), -07122 (SMC no. 11), and -08442 (SMC no. 5). Two genes in SMC no. 20 (NCU05760 and -05769) were only found in *Neurospora* and *Sordaria* genomes. We focused our further investigation on SMC genes that had annotations as to their function and that had knockout mutants available from the Fungal Genetics Stock Center (Table S1), especially for genes likely involved in cell wall synthesis and nutrient responses.

**SMCs exhibited diverse expression patterns during the *N. crassa* life cycle.** Expression of predicted SMC genes was analyzed across four stages of conidial germination and asexual growth, cultured on Bird medium (BM; 79) at 25°C, on Bird medium at 37°C (BM37), on potato dextrose medium (PDA) at 25°C, and on maple sap medium

**TABLE 2** Core genes of secondary metabolism clusters in *N. crassa*

| SMC no. | Chromosome | antiSMASH clusters[a] | JGI[b] | SMC type[c] | CRGS[d] | Notes[e] |
|---|---|---|---|---|---|---|
| 1 | I (7914549–7932266) | NCU00583 to -00587 **(NCU00583)** | Not identified | PKS | BM, PDA, BM37 | NCU00582, -00589 (BM) |
| 2 | I (8432960–8474209) | NCU03000 to -03016 | NCU03010 | NRPS-like | BM, SCM | None |
| 3 | I (88672729–8717074) | NCU16468, -02913 to -02926 | NCU02927 to -02918 | PKS | VM, SCM | NCU02908, -02910, -02930, -02931 (SCM) |
| 4 | II (3937420–3969336) | NCU08402 to -08403, -16586, -08404 to -08409, -16588, -08410 | NCU08404 to -08407 | NRPS-like | BM | NCU08411 to -08413 (BM); NCU08411, -08412 (SCM) |
| 5 | II (4062839–4096852) | NCU08436 to -08443, -08445 **(NCU08442)** | NCU08439 to -08443 | NRPS | No condition | None |
| 6 | IV (1967–36181) | NCU10285, -09635 to -09641, -10572, -09642 | NCU10285, -09635 to -09640 | PKS | BM, BM37, VM | NCU09627 to -09630, -09633, -09634 (BM) |
| 7 | IV (508445–548448) | NCU04860 to -04862, -04865 to -04867 **(NCU04867)** | NCU04865 | PKS | PDA | None |
| 8 | IV (4422480–4439201) | Not identified | NCU07307 to -07308 | PKS-like | All conditions | NCU07310 (BM) |
| 9 | V (1702664–1723869) | NCU03583 to -03585 | Not identified | PKS | PDA, BM37 | NCU03582 (SCM) |
| 10 | V (4020488–4031764) | NCU01423 to -01427 | Not identified | Terpene | PDA | None |
| 11 | VI (59272–103867) | NCU07119 to -07126, 07117 **(NCU07122)** | NCU07119 | NRPS | VM | None |
| 12 | VI (398436–416452) | NCU04692, -17064, -12075, -04694 to -04695 | NCU12075 | DMAT | MSM, BM37 | NCU04699 (BM); NCU04690, -04691, -04697 (SCM) |
| 13 | VI (854581–890466) | NCU04797 to -04806 **(NCU04804)** | Not identified | PKS-like | BM | NCU04796 (BM) |
| 14 | VI (3080467–3107588) | NCU06013, -17123, -06007 to -06012 | NCU06013 | PKS | BM37 | NCU06001, -06005, -06019 (BM) |
| 15 | VI (4070819–4105738) | NCU05007 to -05014 | NCU05011 | PKS | BM, PDA, BM37 | NCU05006, -05015 (SCM) |
| 16 | VI (4116143–4157329) | NCU05005 to -05004, -10597, -05001, -12150 to -12152, -05000, -12154, -04998 to -04996, -10537, -04994, -12156 **(NCU04998)** | NCU05000, -12154 | NRPS-like | No condition | NCU04992 (BM); NCU04991, -04992 (SCM) |
| 17 | VII (1083585–1110519) | NCU08395 to -08399, -12034 | NCU08399 | PKS | BM37 | NCU08390 (BM) |
| 18 | VII (1529996–1574489) | NCU04528 to -04535, -12022 to -12021, -06170 | NCU04531 | NRPS | BM37, SCM | NCU04520 to -04524, -04537 (BM) |
| 19 | VII (1954064–1973561) | NCU06054 to -06056, -10683, -06052 to -06051 | Not identified | Terpene | BM, VM | NCU06057 (BM) |
| 20 | VII (3720911–3763243) | NCU05755 to -05763, -17267, -05764, -17268 to -17271, -05766 to -05769 **(NCU05759)** | Not identified | RiPP[f] | MSM, VM, SCM | NCU05770 (BM); NCU05751, -05752, -05773, -05774 (SCM) |

[a]Orphan genes are listed in parentheses with a bold font.
[b]SMCs reported at the JGI MycoCosm database (Grigoriev et al. [18]).
[c]SMC-type identification follows the glossary in the antiSMASH documentation.
[d]Environmental and developmental conditions in which coordinately regulated gene sets (CRGS) were identified, including asexual growth on BM, MSM, and PDA at 25°C, asexual growth on BM at 37°C (BM37), conidiation on Vogel's medium (VM), and sexual development on SCM at 25°C (as in Fig. S1).
[e]An additional five genes on each side of the predicted SCM were examined to ascertain whether they exhibited the same expression patterns as any two or more of the SCM core genes during asexual growth on BM at 25°C and/or during sexual development on SCM at 25°C.
[f]RiPP, post-translationally modified peptides.

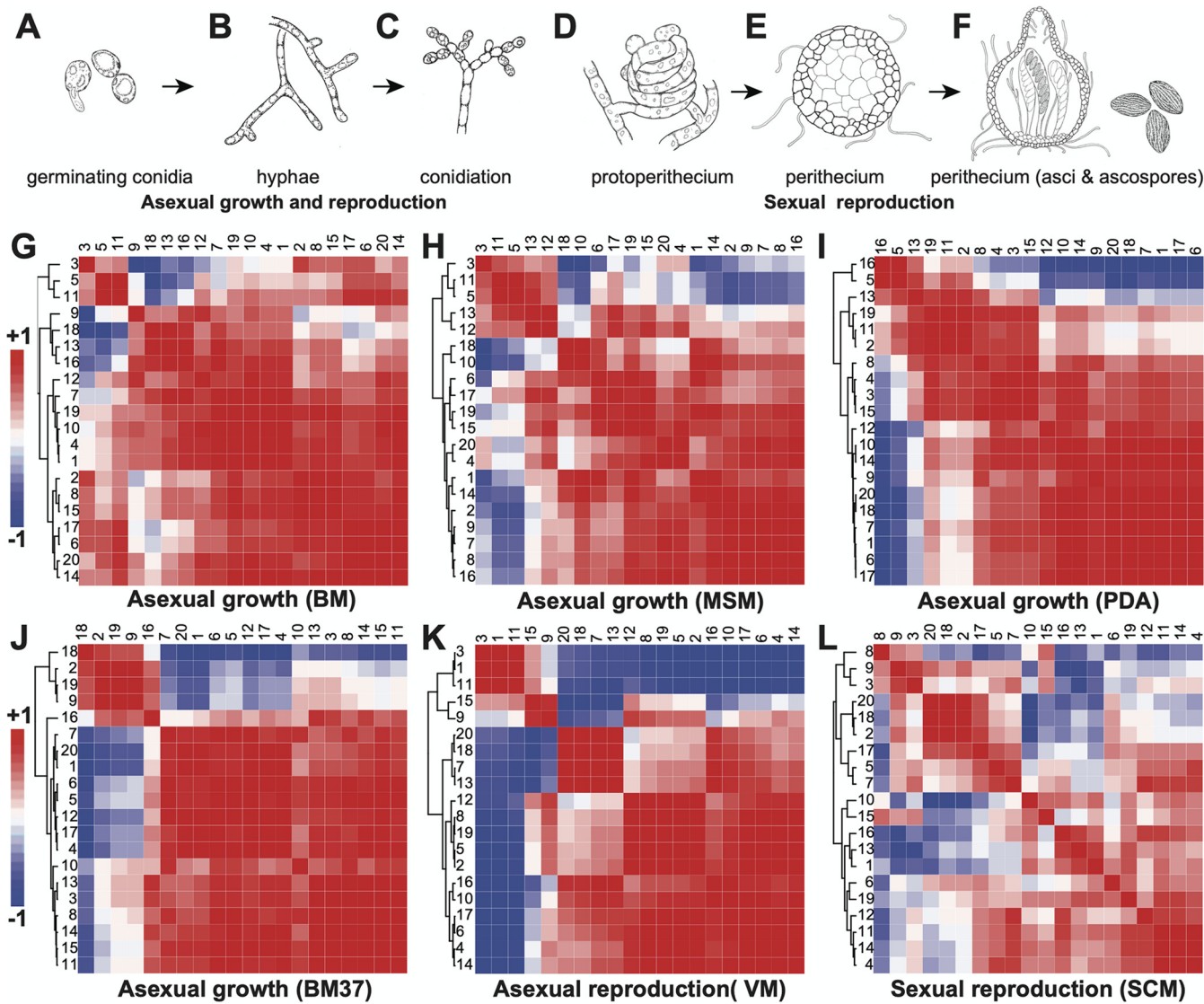

**FIG 1** Morphology and transcriptomics for developmental stages of *Neurospora crassa*. (A to F) Morphologies depicted are (A) germinating conidia, (B) initial hyphal growth, (C) asexual sporulation, (D) initiation of sexual development in protoperithecia, (E) development of perithecia, and (F) sexual sporulation. (G to L) Hierarchically clustered heatmaps of correlation coefficient based on average fold change of gene expression (red, positive; blue, negative) within enumerated secondary metabolic clusters (Table 1) during asexual growth on (G) Bird medium (BM) at 25°C, (H) maple sap medium (MSM) at 25°C, (I) potato dextrose agar medium (PDA) at 25°C, and (J) Bird medium at 37°C; during conidiation on (K) Vogel's medium (VM) at 25°C (from Sun et al. [84]) and during sexual sporulation on (L) synthetic crossing medium (SCM) at 25°C (from Wang et al. [63]).

(MSM; 66) at 25°C. Expression of the SMC genes was also analyzed for cultures on Vogel's medium during asexual sporulation and on synthetic crossing medium (SCM; 80) at 25°C across eight stages of sexual development, starting with the production of protoperithecia (Tables S1 and S2). Expression for neighbor genes of predicted SMCs was also examined for culture on BM at 25°C and on SCM at 25°C across sexual development (Table 2). RNA sequencing data are also available at the GEO database with accession numbers GSE41484, GSE101412, and GSE168995. *N. crassa* SMCs exhibited a dramatic variability in expression regulation depending upon developmental status and upon growth condition (Fig. 1A to F). Interestingly, expression for *N. crassa* SMCs generally followed a general-to-specific trend through the stages of asexual to sexual developments. A large portion of the SMCs exhibited congruent expression patterns during asexual growth, suggesting that these SMCs were likely subjected to similar expression regulation.

The grouping of SMCs by correlation clustering was not consistent among the four asexual growth conditions—especially between the treatments of 25°C and 37°C. In

general, there were two coordinated expression groups among all SMCs that could be easily recognized in all four conditions (Fig. 1G to J). Higher expression correlation of SMCs was observed when *N. crassa* was cultured on BM and MSM at 25°C than when it was cultured on BM at 37°C, with SMCs no. 3, no. 5, and no. 11 or SMCs no. 4, no. 10, no. 14, no. 15, and no. 18 exhibiting two distinct expression patterns, and SMCs no. 4, no. 10, no. 14, no. 15, and no. 18 also exhibited a similar expression pattern on PDA at 37°C (Fig. 1G to I). For asexual growth at 37°C, there were also two distinct types revealed by expression correlation analysis: those that exhibited an upregulation during the production of germ tubes (SMCs no. 2, no. 9, no. 18, and no. 19) and those that exhibited a general downregulation during germ tube production (the other SMCs). The two groups could be differentiated from each other by expression changes occurring at the first step of conidial germination (Fig. S1). The smaller group of only 4 to 10 SMCs exhibited a low but generally upregulated expression from spore to germ tube appearance, implying an activated status. A large group of the other SMCs exhibited downregulated expression before germ tube extension, implying inhibited activity during the developmental time course. The groups composing the small, generally upregulated SMCs and groups composing the large, generally downregulated SMCs were inconsistent in a comparison between 25°C and 37°C. SMCs no. 3, no. 5, and no. 11 were grouped together at 25°C, especially in the BM and MSM experiments. Furthermore, SMC no. 2, no. 9, no. 18, and no. 19 exhibited upregulation during early conidial germination in 37°C experiments, suggesting that temperature affects the activities of these SMCs. *N. crassa* conidial germination was examined here on three media: standard BM, a precisely controlled medium with a sparse range of essential simple nutrients, PDA, a richer medium with a more natural distribution of complex nutrients and carbohydrates, and MSM, a minimal but naturally complex medium. These experiments spanned a range of potential environments experienced by *N. crassa* in its natural state. However, many other potential environments are possible, and none of these media are the fungus' natural habitat. Therefore, additional characterization of expression profiles for SMCs collected from growth on various media, especially those with different carbon and nitrogen ratios, are warranted to clarify the diversity of response to growth medium of SMC expression during the asexual phase.

SMCs manifested two clustered patterns during conidiation. Genes in SMCs no. 1, no. 3, no. 9, no. 11, and no. 15 exhibited similar patterns of expression, and the expression of genes in all other SMCs exhibited another distinct pattern (Fig. 1K). In contrast to their more uniform expression during asexual growth and conidiation, the most discrepant expression patterns among SMCs were observed during sexual development (Fig. 1L): only SMCs no. 2, no. 18, and no. 20 exhibited similar patterns of expression across the process, suggesting greater functional divergence among SMCs during sexual reproduction.

**Coordinately regulated gene sets (CRGS) were identified.** In general, genes within an SMC behaved similarly in one or more of the sampled conditions (Fig. S1). Using the lag-penalized weighted correlation for clustering short time series (78), CRGSs were identified in 17 SMCs—all except SMCs no. 5, no. 16, and no. 11—during developmental stages and processes of the *N. crassa* life cycle (Fig. S1, Table 2, Tables S3 to S5). The CRGSs were supported by at least half of the genes that were classified in the same expression regulation cluster by lag-penalized weighted correlation in an SMC under certain growth/development conditions. Beside SMC no. 8, CRGSs under at least one condition during asexual growth were identified for 15 SMCs: no. 1 (BM25C, BM37C, PDA), no. 2 (BM25C), no. 4 (BM25C), no. 6 (BM25C, MSM), no. 7 (PDA), no. 9 (BM37C, PDA), no. 10 (PDA), no. 12 (BM37C, MSM), no. 13 (BM25C), no. 14 (BM37C), no. 15 (BM25C, BM37C, PDA), no. 17 (BM37C), no. 18 (BM37C), no. 19 (BM25C), and no. 20 (MSM). CRGSs were identified for five SMCs (no. 3, no. 6, no. 11, no. 19, and no. 20) during asexual reproduction of mitotic conidia (conidiation), and CRGSs were identified during sexual reproduction of meiotic ascospores for four SMCs (no. 2, no. 3, no. 18, and no. 20). Coordinated expression of the core genes in an SMC under specific

developmental and environmental conditions indicates relevant regulation and activity of the SMC under those conditions.

**SMCs are actively regulated during asexual growth.** Asexual growth was sampled in four growth conditions: cultures on BM, MSM, and PDA at 25°C, as well as culture on BM at 37°C. Among the media used here, BM is specifically designed for asexual growth and reproduction only. In general, expression of SMCs exhibited a w-shaped pattern: most SMC genes were downregulated during the establishment of polar growth from fresh conidia to the appearance of the germ tube, then upregulated during the appearance of the first hyphal branch (Fig. S1). Of the 20 SMCs identified, 15 (SMC no. 1, no. 2, no. 4, no. 6 no. 7, no. 8, no. 9, no. 10, no. 12, no. 13, no. 14, no. 15, no. 18, no. 19, and no. 20) were identified with CRGSs in at least one of the four asexual growth conditions (Fig. S1). SMC no. 1, no. 8, no. 9, no. 13, and no. 20 exhibit differential expression dynamics during asexual growth. (Fig. 2).

SMC no. 1 comprised five genes, including three annotated genes coding for mitogen-activated protein kinase *os-5* (NCU00587), nonanchored cell wall protein *ncw-6* (NCU00586), and phytoene synthase *al-2* (NCU00585). Except for the flanking open reading frame (ORF) NCU00583, the ORFs of SMC no. 1 were coordinately expressed across conidial germination cultured on BM at both 25°C and 37°C (Fig. 2A to D, Fig. S1). Cultured on BM—which supports only asexual growth—*N. crassa* genes in SMC no. 1 other than NCU00583 exhibited downregulation of expression before germ tube appearance, followed by slight upregulation of expression during germ tube extension. Conidia germinating on PDA exhibited upregulation of expression of genes in SMC no. 1, other than NCU00583 and -00586. Genes NCU07307 and -07308 in SMC no. 8, encoding two subunits (*cel-1* and *cel-2*) of a fatty acid synthase complex, exhibit a very highly coordinated expression (Fig. 2E to H). This high consistency with expected coordinated regulation across different experimental settings for tightly associated genes (subunits) represents an internal validation of the expression data presented (c.f. subunits of the proteasome in Townsend et al. [81]).

Of the three genes identified as belonging to SMC no. 9, a coordinated upregulation of expression after the germination of conidia was observed for a hypothetical protein-coding gene (NCU03583) and a perithecial gene (NCU03584, *per-1*) in Bird cultures, PDA cultures, and during conidiation (Fig. 2I to L). These two genes exhibited no significant expression changes when *N. crassa* was cultured at 37°C, a temperature that inhibits the sexual development in *N. crassa*. NCU03584 (*per-1*) is a polyketide synthase that is involved in melanin synthesis in *N. crassa* (63). Its name, *per-1*, derives from a phenotype of production of perithecia that fail to produce ascospores (82, 83).

SMC no. 13, a secondary-metabolic cluster newly identified in *N. crassa* with antiSMASH, includes seven genes encoding fructose-1,6-bisphosphatase (*fbp-1*, NCU04797), alpha-1,2-mannosidase (*gh92-1*, NCU04798), polyadenylate-binding protein (*pabp*, NCU04799), ubiquitin-10 (*ubi-10*, NCU04800), chalcone synthase (*csy*, NCU04801), peroxisomal biogenesis factor (*pex11*, NCU04802), and nitropropane dioxygenzse-2 (*npd-2*, NCU04803) and three hypothetical protein-coding genes (NCU04804 to -04806). All of these genes exhibited dynamic regulation during germination of conidia (Fig. 2M to P). Expression was high in activated conidiospores and was coordinately downregulated toward germination for cultures on Bird medium with simple sugar as the sole carbon resource (Fig. 2M to P). However, during conidial germination on MSM and PDA media featuring complex carbohydrates, these genes were generally (but less coordinately) consistent in expression level or were upregulated (Fig. 2N and O). Therefore, the roles of these genes as a cluster in response to ambient carbon resources in natural environments of *N. crassa* warrant further investigation.

SMC no. 20 is a large SMC to which 19 genes were assigned, 7 of which are functionally annotated, coding for peptidase M4 *mpr-20* (NCU05756), nitrilase-5 *nse-5* (NCU05757), pheromone receptor *pre-2* (NCU05758), oligopeptide transporter-5 *opt-5* (NCU17269), zinc-finger transcription factor-10 *znf-10* (NCU05767), mating-response protein *poi-2* (NCU05768), and a putative septal pore-associated protein (NCU05769).

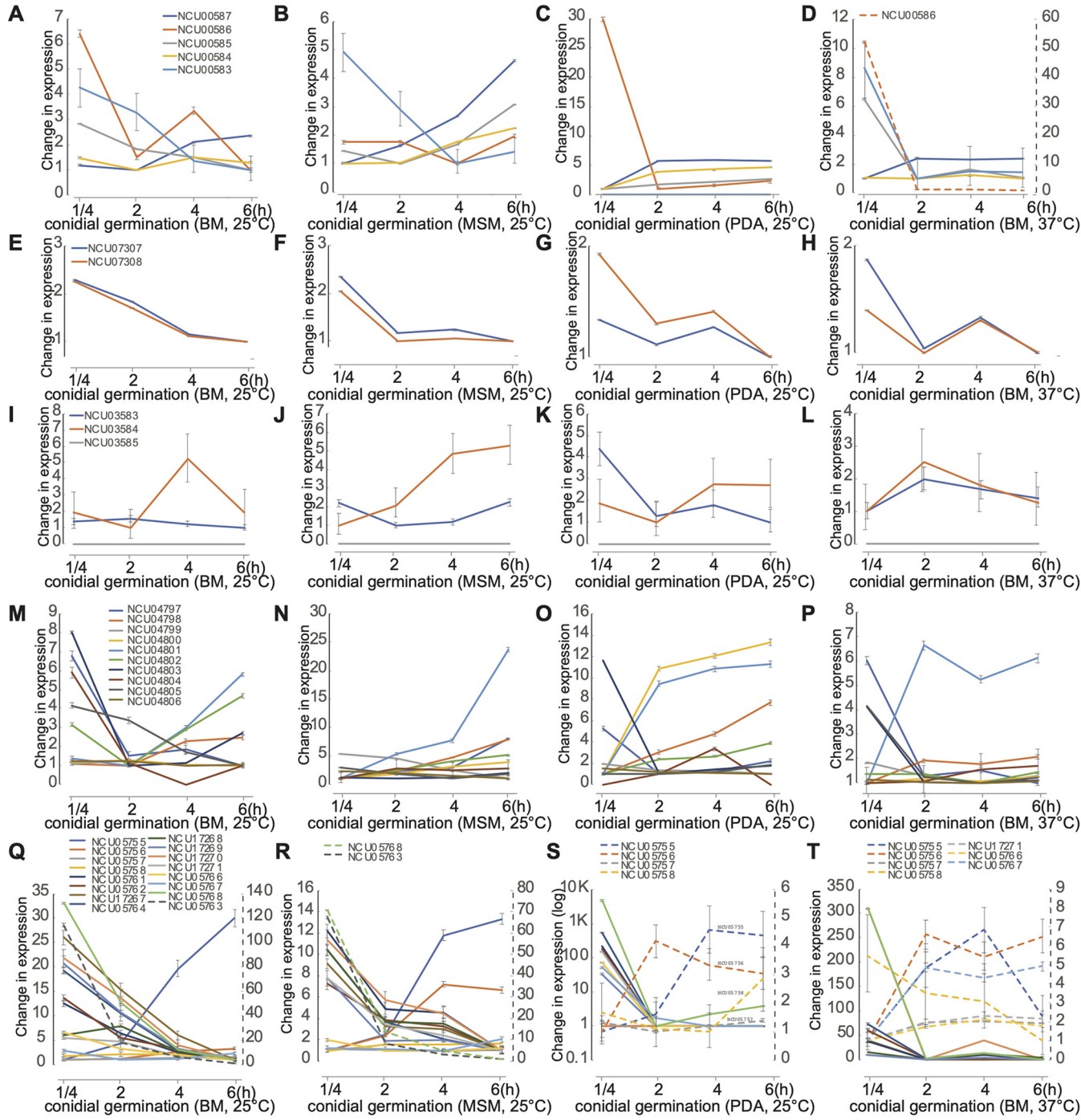

**FIG 2** LOX expression profiles for selected secondary metabolic clusters (SMCs) no. 1, no. 8, no. 9, no. 13, and no. 20 during key stages of asexual growth (germination of conidia to the first hyphal branching on Bird medium (BM), maple sap medium (MSM), and potato dextrose agar medium (PDA) at 25°C and on Bird medium at 37°C; Table S1). (A to T) Expression profiles for genes in (A to D) SMC no. 1, (E to H) SMC no. 8, (I to L) SMC no. 9, (M to P) SMC no. 13, and (Q to T) SMC no. 20. Profiles for each gene are color-coded. Expression levels of some genes (dashed lines) are quantified by the secondary right-hand dashed *y* axis. Whiskers represent 95% credible intervals.

Excluding three cluster-flanking genes (NCU05755, -05756, and -05757) and two genes exhibiting no measurable expression (NCU05759 and -05760) that were identified only in the *Neurospora* and *Sordaria* genomes, the remaining 14 genes in the cluster exhibited coordinated expression during conidial germination at 25°C (Fig. 2R and S). These genes exhibited relatively lower expression fold changes and were less coordinated in expression when cultured at 37°C (Fig. 2T). Expression of genes in SMC no. 20 was highly coordinately regulated during sexual development (discussed below).

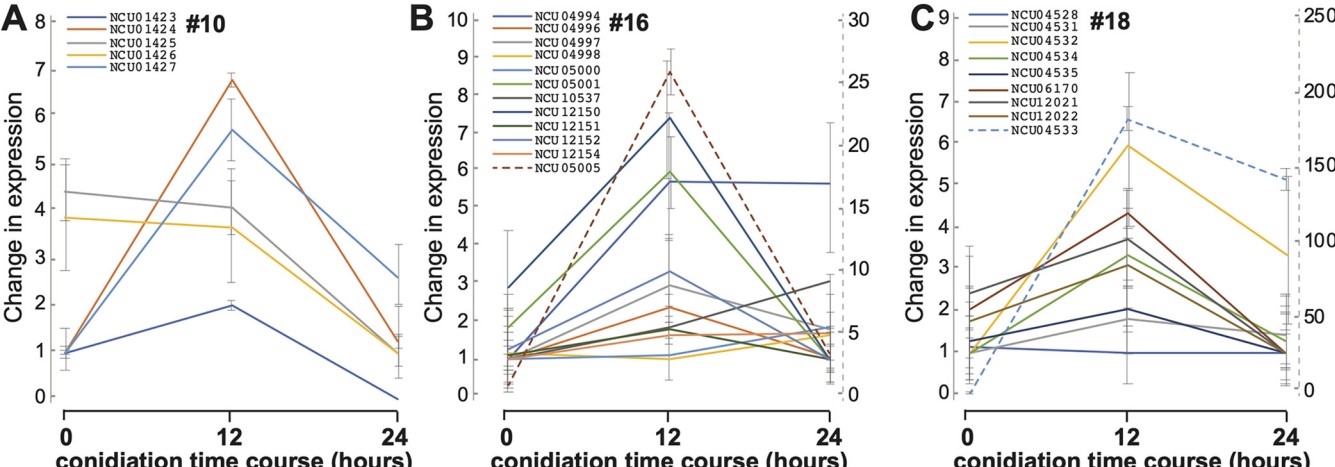

**FIG 3** Expression profiles exhibited similarly highly coordinated expression among genes for three SMCs during conidiation on Vogel's medium (see also Table S1). (A to C) Expression profiles relative to time points 0, 12, and 24 h of culture on Vogel's medium for (A) SMC no. 10, (B) SMC no. 16, and (C) SMC no. 18. Line plots are color-coded by gene. Expression levels of some genes (dashed lines) are quantified by the secondary right-hand dashed *y* axis tick labels. Whiskers indicate 95% credible intervals.

**SMCs are actively regulated during conidiation.** Three data points representing three stages on Vogel's medium 0, 12, and 24 h after exposure to light to induce conidiation (published in reference 84) were reanalyzed. Conidia formed 12 h after exposure to light, and abundant conidia had been produced by 24 h subsequent to inoculation (65). High-rate production of conidia that forms a conidiation band in *N. crassa* is rhythmic, with a period of 10 to 11 h at 25°C. Thus, the peak production rate of conidia occurs about 12 h after inoculation of mycelia (85). A long history of genetic analysis of *N. crassa* conidiation has identified many key genes in the sporulation pathway (86–91). Based on these three data points, the SMCs exhibited more diverse regulation patterns during conidiation than during the conidial germination and asexual growth, including continual up- and downregulation as well as peaking expression patterns reaching a peak or nadir at the 12-h midpoint (Fig. S1). In the first three time points during conidiation, genes in SMCs no. 10, no. 16, and no. 18 exhibited high levels of coordination of expression during *N. crassa* conidiation (Fig. 3A to C).

SMC no. 10 includes five genes, NCU01423 to -01427. NCU01427 is annotated as geranylgeranyl pyrophosphate synthase *al-3*, which is light-regulated and involved in carotenoid accumulation during conidiation. It has a highly analogous function to that of *al-2* in SMC no. 1 (63, 92, 93). Genes of SMC no. 10 exhibited a coordinated upregulation after a short 15-min exposure to light, based on data from a previous study (94). SMC no. 16 comprises 15 genes, including those annotated as leucine aminopeptidase-2 *lap-2* (NCU04994), xylanase *gh10-3* (NCU04997), ent-kaurene oxidase *ci-1* (NCU05001), sugar transporter-29 *sut-29* (NCU12154), and transport of metals-61 *trm-61* (NCU12156). SMC no. 18 has 10 genes, including those annotated as laccase *lacc* (NCU04528), nuclear migration protein *nmp-1* (NCU04534), and spindle pole body component *div-19* (NCU04535). Both *nmp-1* and *div-19* are involved in regulation of microtubule activities and mitotic spindle establishment, which are critical for conidial germination, vegetative growth, and conidiation (95, 96).

**SMCs appear to have roles in the asexual-sexual switch.** Although the asexual-sexual switch is a critical decision-point in the life cycles of many fungi, it has not been well studied (66). One of the reasons studying the asexual-sexual switch in *N. crassa* has been challenging is that it has been difficult to knowledgeably mimic natural conditions for the decision and therefore difficult to properly acquire representative samples in the laboratory. The switch was not directly sampled in the experiments discussed here. Ideally, fine-scaled sampling over this 5- to 7-day process on a medium with similar carbon and nitrogen levels as the fungus' natural habitats, along with statistical approaches to quantify heterogeneous components, will be supportive of a comprehensive understanding of this

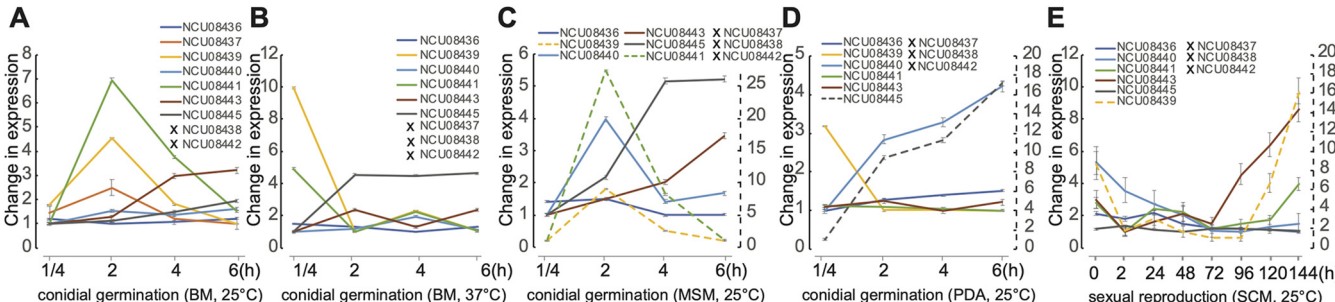

**FIG 4** Expression profiles for SMC no. 5 during asexual growth and sexual reproduction of *Neurospora crassa*. (A to E) *N. crassa* was cultured and gene expression measured on (A) Bird medium at 25°C, (B) Bird medium at 37°C, (C) maple sap medium at 25°C, (D) PDA medium at 25°C, and (E) synthetic complete medium (SCM) at 25°C across eight stages of sexual reproduction from protoperithecia: 2 h after crossing and 24, 48, 72, 96, 120, and 144 h of perithecial development toward the maturation of ascospores (63). Line plots are color-coded by gene. Expression levels of some genes (dashed lines) are quantified by the secondary right-hand dashed *y* axis tick labels. Whiskers indicate 95% credible intervals. Genes with no detectable expression during the sampled processes are designated by a bold X in the legend.

process. We revealed that some genes in SMC no. 5 exhibit likely roles in the switch. No CRGSs were identified in SMC no. 5 (leptomycin-B resistance protein *pmd-1*, NCU08439, plus NCU08440 to -08443). However, many genes in this cluster were upregulated during conidial germination near the stage at which branching hyphae appeared (Fig. 4A to D). Genes in this cluster also exhibited coordinated expression on synthetic crossing medium during the early stages of sexual development, from protoperithecia to 24 h after crossing, and were downregulated after crossing during sexual development (Fig. 4E). Such a congruent peak and fall suggest that these genes take on active roles in germinating conidia, and in protoperithecia before crossing, these stages represent the start and the end of the asexual-sexual switch. Therefore, the roles of SMC no. 5 in the asexual-sexual switch are of special interest.

Within SMC no. 5, the leptomycin-B resistance protein PMD-1 (NCU08439) is well studied. It is structurally similar to Ste6p in *Saccharomyces cerevisiae*, an ortholog of PMD-1 that plays an unclear role as a transmembrane transporter of the mating factor. *N. crassa* PMD-1 and *S. cerevisiae* Ste6p are orthologs of mating pheromone factor (p-factor) or mating factor (m-factor) in *Schizosaccharomyces pombe* (97, 98). Knockout mutants of *pmd-1* have been reported with abnormal growth and reduced pigment production during asexual growth and abortive protoperithecia that failed to be fertilized and to produce meiotic spores (54, 99), a phenotype that we confirmed in this study. However, *pmd-1* exhibited differential expression between low-carbon MSM and carbon-enriched PDA medium. The *pmd-1* gene product is a target of numerous antifungal drugs, such as leptomycin B, and further investigation of the response of *pmd-1* to the environmental carbon level could be helpful in understanding the roles of PMD-1 in fungal defense.

**SMCs are actively regulated during sexual growth and development.** Sexual reproduction in *N. crassa* starts with development of protoperithecia which protect the female receptive hyphae and then, after fertilization, mature into perithecia, within which resistant ascospores are produced and ultimately dispersed (53, 68). Secondary metabolites are known to be associated with sexual development in *N. crassa* (100, 101). A recent study reported that the production of insecticidal neurosporin A counteracted feeding attacks by arthropods during sexual development of *N. crassa*. Neurosporin A was shown to be a product of polyketide synthase gene cluster 6 (*pks-6*), homologous with the highly reducing polyketide synthase gene cluster from the *Magnaporthe oryzae* genome (59, 60).

Expression of genes within SMCs was more divergent during sexual development. Nevertheless, a general pattern of downregulation from protoperithecia to postcrossing perithecia was highly evident (Fig. S1). For many SMC genes, peak expression during ascospore development occurred between 48 and 96 h after crossing. CRGSs during sexual reproduction of meiotic ascospores were identified for four SMCs, no. 2, no. 3, no. 18, and no. 20 (Fig. 5A to D), within which only SMC no. 3 and SMC no. 20 (addressed above)

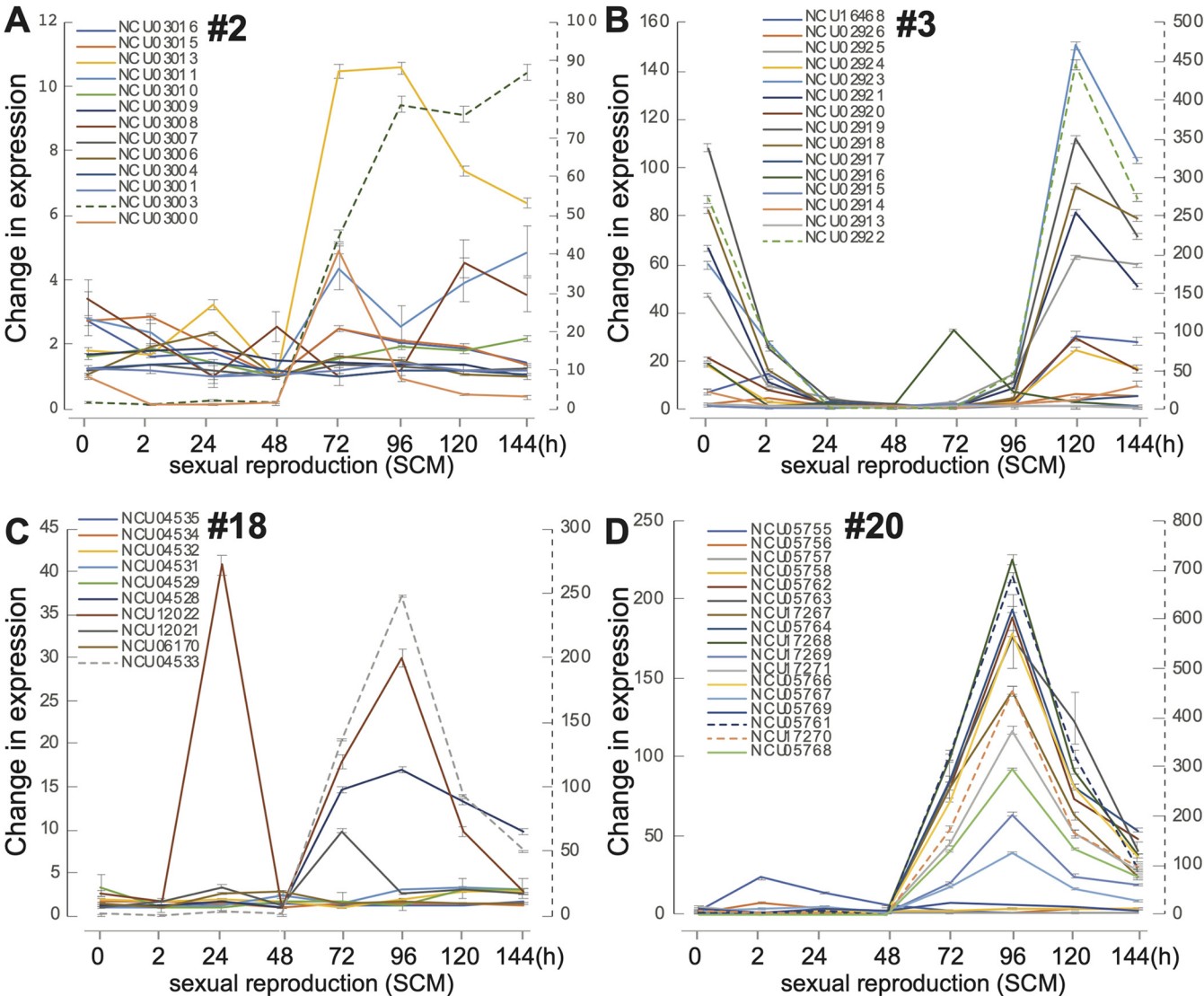

**FIG 5** Expression profiles for selected secondary metabolic clusters (SMCs) during *Neurospora crassa* sexual reproduction cultured at 25°C on synthetic complete medium (SCM; Table S1). (A to D) Expression profiles for genes in (A) SMC no. 2, (B) SMC no. 3, (C) SMC no. 18, and (D) SMC no. 20. *N. crassa* was sampled at eight stages of sexual reproduction from protoperithecia, 2 h after crossing, and 24, 48, 72, 96, 120, and 144 h of perithecial development toward the maturation of ascospores (63). Line plots are color-coded by gene. Expression levels of some genes (dashed lines) are quantified by the secondary right-hand dashed *y* axis tick labels. Whiskers indicate 95% credible intervals.

genes exhibited coordinated expression during sexual development from protoperithecia to mature ascospores (Fig. 5B and D). SMC no. 3 has over seven genes (NCU02918 to -02926), including fruiting body maturation-1 (NCU02925, *fbm-1*) and three possible polyketide synthases (NCU02918 *pks-6*, -02919, -02923), as well as epoxide hydrolase-1 (NCU02924, *eph-1*). These genes exhibited highly coordinated downregulation during the early stages of sexual reproduction before meiosis and then coordinated upregulation toward the maturation of the asci and ascospores. The cluster featuring *pks-6*—regulating the production of the insecticidal polyketide neurosporin A—is part of SMC no. 3. However, assessed knockouts of the *pks*-like genes in SMC no. 3 did not manifest phenotypes in either asexual or sexual development. Therefore, a further biochemical assay would be warranted, investigating the effects of this gene knockout on the production of neurosporin A.

**Temperature and light affect the expression of genes in SMCs during asexual growth.** Environmental factors, especially environmental stress from exposure to intensive light and/or extreme temperature, play critical roles in regulating SMC activities

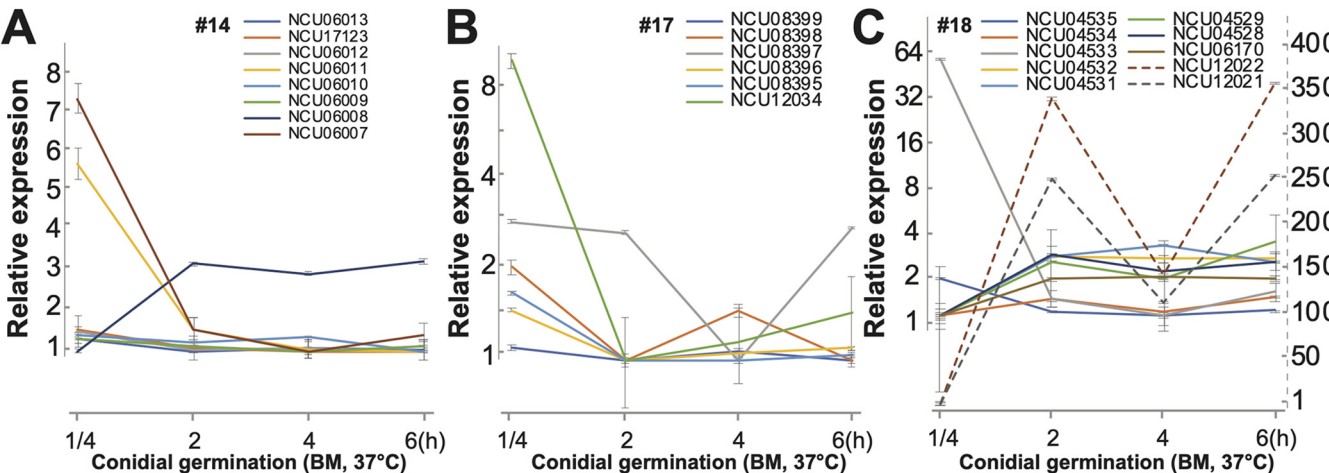

**FIG 6** Expression estimates and 95% confidence intervals of genes at four stages of *Neurospora crassa* conidial germination on Bird medium at 37°C: fresh spores, polar growth, germ tube extension, and first hyphal branching. (A to C) Expression of genes in (A) SMC no. 14, (B) SMC no. 17, and (C) SMC no. 18. The secondary right-hand dashed *y* axis tick labels quantify expression of neighboring genes NCU12021 and -12022, which exhibited much larger changes across conidial germination than the other genes in the cluster. Line plots are color-coded by gene. Expression levels of some genes (dashed lines) are quantified by the secondary right-hand dashed *y* axis tick labels. Whiskers indicate 95% credible intervals.

(29, 30). *N. crassa* grows with normal morphology and biology between 15°C and 37°C. Faster growth and branching were observed in higher temperatures, with a peak growth rate at 35°C (102, 103). In three SMCs (no. 14, no. 17, and no. 18), CRGSs were detected only for BM37C cultures within the four asexual growth experiments (Fig. S1). These three SMCs are representative of the three expression correlation patterns in BM37C (Fig. 6). Some genes in SMC no. 14 were annotated, including NCU06009, -06010, -06011, and -06013, encoding oxidoreductase, mutanase, multidrug-resistance protein 3, and polyketide synthase 1, respectively. All four of these genes exhibited decreasing expression following commencement of the conidial germination process. Similar downregulation was also observed for genes in SMC no. 17, including genes coding for pre-mRNA splicing factor DIM-1 (NCU08395), aldose epimerase-2 (NCU08398), and polyketide synthase 4 (NCU08399). More dynamic expression was observed for SMC no. 18 genes, including spindle-pole body component gene (NCU04535) and NudF-2 (NCU04534), genes that are required for nuclear migration during development (55). CRGSs identified in MSC no. 18 also suggested highly coordinated regulation among these genes during sexual development.

There have been many temperature-sensitive mutants reported in *N. crassa*, some of which exhibit abnormal growth in culture at 30°C or higher due to disrupted molecule transport systems and disrupted lipid and fatty-acid synthesis. *Chaetomium globosum*—a close relative to species in the genus *Neurospora*—produces chaetoglobosin, which is toxic to mammals and HeLa cells (104). Chaetoglobosin biosynthesis in *C. globosum* requires the PKS gene *pks-1* (CHGG_00542; 105), which is homologous to one of the three genes in SMC no. 9: the unnamed ORF NCU03584. Expressions of NCU03584 and -03583 were highly correlated when cultured on BM37C and on PDA. No metabolites from *N. crassa* are known to be toxic to mammalian cells, and *Neurospora* is consumed by humans as the fermented legume cake, oncom.

Gene expression was monitored for *N. crassa* cultured in the dark and with 4 h of light exposure (62, 94). Detection of the effects of light exposure on SMC genes required us to revisit previous data from Wu et al. (62). In these experiments, genome-wide gene expression was sampled for *N. crassa* on Bird medium with 2% sucrose under five durations of light exposure, including 0 min (continuous 24-h darkness) and 15, 60, 120, and 240 min of exposure. Out of 177 SMC genes, however, 74 (mostly in SMC no. 3, no. 4, no. 7, no. 9, no. 12, and no. 20) were either silent or without measurable expression over more than 2 sample points. The cultures were kept in the dark for 24 h before light exposure. Therefore, the activities of some SMCs may have started

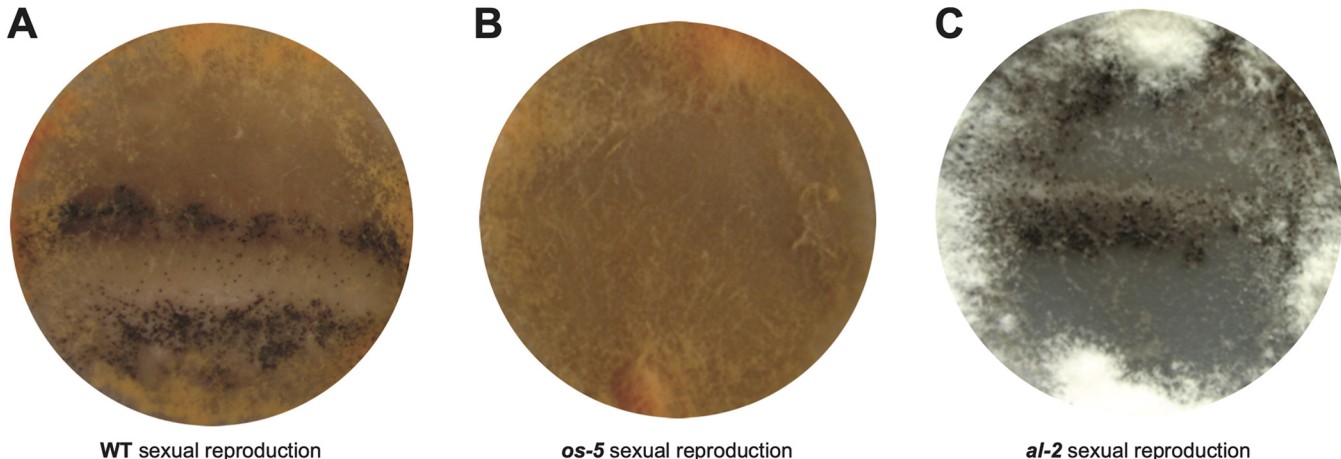

**FIG 7** Normal and knockout phenotypes of some SMC genes. (A) Normal sexual development of wild-type strains (FGSC2489 *mat A* × FGSC1400 *mat a*) on SCM featured orange-colored conidia and dark-colored perithecia along the crossing line. (B) Knockout mutants (FGSC1638 *mat A* × FGSC18203 *mat a*) of *os-5* exhibited no sexual development on SCM. (C) Knockout mutants (FGSC17611 *mat A* × FGSC799 *mat a*) of *al-2* on SCM featured white hyphae and conidia.

completely inhibited and may not have had sufficient time to respond during that short period of light exposure. More data collected from fungal growth under longer light exposures could help to understand why such a large portion of SMC genes were detected as inactive in this experiment.

**Some SMC genes exhibited phenotypes when knocked out.** Knockout mutants of 126 SMCs genes from a previous systematic knockout project (56) were systematically examined for phenotypes in both asexual and sexual growth (Fig. 7, Table S1). The genes *al-2* and *al-3* exhibited phenotypes in asexual growth and conidiation, whereas *ada-6*, *os-5*, *poi-2*, and *pmd-1* exhibited phenotypes in sexual development. Unlike the wild type (Fig. 7A), Δ*os-5* (FGSC1638 *mat A* × FGSC18203 *mat a*) produced no sexual structure or meiotic spores (Fig. 7B). Δ*al-2* (FGSC17611, *mat A*) produced white-colored hyphae and conidia and normal protoperithecia, which further developed into normal perithecia after crossing with the *mat a* wild-type strain (Fig. 7C).

The high expression levels of SMC no. 20 genes *pre-2* and *poi-2* in conidia and protoperithecia are consistent with predicted functions that are associated with the mating process. These two genes are involved in mating and the initiation of sexual development: knockout mutants of *pre-2* in a *mat-a* strain produce protoperithecia that cannot be fertilized by conidia from the opposite mating type to form perithecia, while knockout mutants of *pre-2* in a *mat-A* strain can reproduce sexually, producing viable meiotic spores (106). All *poi-2* mutants were reported with defects in conidial fertility and the mating response, including reduced vegetative growth, few protoperithecia, as well as low viability of their sexual progeny (107). These phenotypes are highly consistent with *pre-2* and *poi-2* expression profiles. Other genes in SMC no. 20 were highly coordinately expressed with *pre-2* and *poi-2*. However, no knockout phenotypes were observed for any other gene in SMC no. 20. Therefore, an integrative understanding of these mostly unannotated genes requires further molecular and functional analysis that could reveal crucial details of the regulatory networks of *pre-2* and *poi-2*.

**Orphan genes disrupt synteny and challenge functional prediction of SMCs.** Synteny is the key character applied to perform prediction of fungal SMCs. Expression profiles largely support the SMCs predicted by antiSMASH. However, genes in the flanks often behave differently from the centrally located genes in the clusters. Therefore, we expanded searches for the outer boundary of the SMCs—beyond the centrally located core genes—and often identified additional neighbor genes with a similar expression regulation with the centrally located genes (Table 2).

Synteny was observed among these neighbor genes and among their orthologs in other fungal genomes. Synteny was likely maintained by natural selection for conserved,

coregulated function. For example, genes NCU05750 to -5753 and NCU05770 near SMC no. 20 exhibited similar expression patterns during asexual growth and sexual development (Table S1). Nevertheless, a few genes within the predicted SMCs are "orphan" genes that have no homologs in a representative set of ascomycetous genomes (108). Furthermore, knockouts of orphan genes without homologs in other fungal genomes did not exhibit any observable phenotypes under any of the laboratory conditions examined. Therefore, the presence of the orphan genes in conserved SMCs is an evolutionary genomic conundrum worthy of further investigation. Perhaps relocation of orphan genes is not infrequent, although the mechanisms underlying such relocations are not well understood (108, 109). The presence of these orphan or young genes has previously confounded the identification of fungal SMCs (110). Orphans that were predicted in SMCs by antiSMASH could simply be due to their being embraced by syntenic orderly conserved SMC genes within the cluster. These orphan genes were inactive in most of the stages sampled along the *N. crassa* life cycle, suggesting that their functions are nonessential in laboratory environments.

**Conclusion.** In this study, we verified 14 SMCs that had been predicted in the JGI database and identified an additional 6 SMCs, distinguishing their core genes in the genetic model *Neurospora crassa*. Furthermore, identified stages in the life cycle in which the SMC genes are actively expressed and dynamically regulated disclosed possible roles of some SMCs in asexual and sexual development, especially along with knockout phenotype evidence for the SMCs that exhibited divergent expression regulation during sexual reproduction. Coordinately regulated gene sets (CRGSs) were identified in 18 predicted SMCs. Divergent activity of SMCs between asexual and sexual development was observed— including upregulation of carotenoid derivatives in asexual growth and upregulation of the mycotoxin neurosporin A gene cluster in sexual development. Phenotypes were systematically investigated for knockout mutants available in 126 SMC genes. Except for a few knockout mutants that produced striking phenotypes in asexual and sexual growth, most SMC gene knockouts exhibited wild-type phenotypes across the *N. crassa* life cycle on artificial media and under laboratory conditions. This lack of phenotypic effect calls for increasingly thorough investigations of environmental conditions that may reveal knockout phenotypes.

We also reported the presence of orphan genes in SMCs, especially at the flank regions. We suggested that further investigation of the regulation and function of the orphan genes is needed. Further investigation could elucidate the relevant evolutionary roles and ecological functions—including but not limited to mycotoxins, antibiotics, biopolymers, and biohazards—of fungal SMCs. Our observations raise several questions deserving further investigation regarding SMC gene functional integration and SMC definition in *N. crassa*. First, we have demonstrated divergent regulation of SMCs in response to different ecological and/or developmental conditions. However, how the natural environmental, spatial, and temporal conditions for expression of genes in an SMC are associated with their functions in secondary metabolism requires more comprehensive examination. Second, we have confirmed that expression of genes in SMCs is frequently coordinately regulated and, interestingly, in different patterns under different conditions. Whether the complexity of differential coordination in distinct developmental processes arises from one, a few, or many genetic regulatory factors is a challenge for future investigations. Third, we have discovered uncoordinated expression of some genes within predicted SMCs, including both flanking genes and centrally located genes. These observations provide support to efforts to include transcriptomics in improved algorithms for prediction of SMCs. Last, our research has revealed nonsyntenic and nonconserved elements in many SMCs by comparison to diverse fungal genomes, raising new questions regarding the means by which gene expression evolves in clusters of coregulation. Fungal SMCs present a tractable subset of the genome for investigation of these and many other essential questions of molecular evolutionary biology.

## MATERIALS AND METHODS

**Strains and culture conditions.** Asexual growth was sampled for four key stages of conidial germination as in our previous study on this process in *N. crassa* on Bird medium and maple sap medium at

25°C (66). The experiments were performed with *N. crassa mat A* (FGSC2489) macroconidia, which were harvested from 5-day cultures on BM. Then, $1 \times 10^5$ spores were placed onto the surface of a cellophane-covered medium in petri dishes (60 mm, Falcon, reference [ref.] 351007).

Two additional media and one high-temperature condition were investigated—Bird medium (BM, 37°C) and potato dextrose agar (PDA, 25°C; Table S1). Conidia were spread on both media and incubated in a refrigerated incubator (VWR Signature diurnal growth chamber), maintained under continuous white light. Cellophane membranes with fungal tissues were collected when the majority of active spores (>50%) were at each of the following four stages: (i) fresh spores, (ii) first evidence of polar growth, (iii) doubling of the longest axis length, and (iv) commencement of the first hyphal branching (roughly 15, 60, 120, and 240 min, respectively, for cultures on BM at 37°C; and 15, 120, 240, and 360 min, respectively, for cultures on PDA at 25°C). Tissue samples were flash-frozen in liquid nitrogen and stored at −80°C. Biological replicates included all tissues collected from multiple plates in one collection process. Three biological replicates were prepared for each sampled point.

**Identification of SMCs and coexpression SMC gene clusters in *Neurospora crassa*.** SMCs analyzed in this study were obtained by two methods. First, known SMCs in *N. crassa*, *N. discreta*, and *N. tetrasperma* annotated in the JGI Fungal Genome database (18) were compared, focusing on the synteny within each predicted SMC. Second, to predict SMCs in the *N. crassa* genome, the online antiSMASH v5.0 fungal version (76) was executed. This software was developed for rapid genome-wide identification of SMCs and for annotation of bacterial and fungal genomes. The detection stringency of antiSMASH was set to be relaxed, with extra features including known cluster BLAST, subcluster BLAST, cluster Pfam analysis, active site finder, and RREfinder, as well as cluster border prediction based on transcription binding sites.

Expression was profiled for genes identified in the SMCs under the following developmental stages and conditions: (i) four stages of asexual growth (BM, MSM, PDA, and BM37), (ii) three stages of the conidiation, and (iii) eight stages from the protoperithecia to the production of mature ascospores during sexual reproduction and development. Expression correlation among SMCs was calculated using R. Average expression fold changes of genes in each SMC were calculated, and pairwise distance was estimated among all SMCs. Pearson correlation coefficients were then calculated, and expression correlations were used to cluster SMCs with similar regulation. Data from different experiments were analyzed separately; no expression data sets from different experimental settings were combined.

To identify coexpression gene clusters as coordinately regulated gene sets (CRGSs) that were not perfectly synchronized from short time-series data, fold change profiles were analyzed in R using distance-based clustering lag-penalized weighted correlation (78), for which the maximum lag was set to one interval between collected data points, the penalty was set as low, and 10 iterations were performed. Genes with no detectable expression (0 mapped reads) in more than two time points were excluded from analysis. To identify up to eight CRGSs per experimental condition, the Cutree function was executed. Genes sharing the same CRGS were considered to be coordinately regulated. SMCs with at least half of their genes listed in the same CRGS under certain growth conditions and developmental processes were considered to be active under those growth and development conditions.

**RNA isolation and transcriptome profiling, data acquisition, and analysis.** Total RNA was extracted from homogenized tissue with TRI Reagent (Molecular Research Center) as in Clark et al. (111), and sample preparation and sequencing followed our previous work (29, 66, 68). Briefly, mRNA was purified using Dynabeads oligo(dT) magnetic separation (Invitrogen). For RNAseq library prep, mRNA was purified from approximately 200 ng of total RNA with oligo(dT) beads and sheared by incubation at 94°C in the presence of Mg (Roche KAPA mRNA HyperPrep, catalogue no. KR1352).

Following first-strand cDNA synthesis with random primers, second-strand synthesis and A-tailing were performed with dUTP to generate strand-specific sequencing libraries. Adapter ligations with 3′ dTMP overhangs were ligated to library insert fragments. Fragments carrying the appropriate adapter sequences at both ends were amplified; second strands marked with dUTP were not amplified. Indexed libraries were quantified by reverse transcription-quantitative PCR (qRT-PCR) using a commercially available kit (Roche KAPA Biosystems, catalogue no. KK4854). Samples with a yield of ≥0.5 ng/$\mu$L and a size distribution of 150 to 300 bp were deemed appropriate for further quality checking and sequencing. The quality of cDNA samples was verified with a bioanalyzer (Agilent Technologies 2100).

The cDNA samples were sequenced at the Yale Center for Genomics Analysis (YCGA). A total of 24 sequencing libraries (3 replicates per condition) were produced with the Illumina TruSeq stranded protocol. The libraries underwent 101-bp paired-end sequencing using an Illumina NovaSeq 6000 (S4 flow cell) according to Illumina protocols. Adapter sequences, empty reads, and low-quality sequences were removed. For each read, the first six nucleotides and the terminal nucleotides at the point where the Phred score of an examined base fell below 20 were trimmed for each read using in-house scripts. If, after trimming, the read was shorter than 45 bp, then the whole read was discarded.

Trimmed reads were aligned to the *N. crassa* OR74A v12 genome from the Broad Institute (55) using HISAT2 v2.1, indicating that reads correspond to the reverse complement of the transcripts and reporting alignments tailored for transcript assemblers. Alignments with a quality score below 20 were excluded from further analysis. Reads were counted for each gene with StringTie v1.3.3 and the Python script prepDE.py provided in the package. StringTie was limited to report reads that matched the reference annotation. Sequence data and experiment details were made available at the GEO database (https://www.ncbi.nlm.nih.gov/geo/).

Statistical analysis of the sequenced cDNA tallies for each sample was performed with LOX v1.6 (77), ignoring raw reads that mapped ambiguously or to multiple loci. Average expression fold changes of genes in each SMC were calculated. Pairwise distance was estimated among all SMCs. Pearson

correlation coefficients were then calculated, and then expression correlations were used to hierarchically cluster SMCs with similar regulation in the form of a heatmap.

Data from a recent study on conidiation and stress response in *N. crassa* (84) were also reanalyzed using LOX to reveal SMC activities during conidiation in the wild-type strain (FGSC4200). In that study, conidiating fungal tissues were cultured on Vogel's medium (112) under constant light at 28°C and sampled three times over a period of 24 h.

**Knockout strains and phenotype identification.** Knockout (KO) strains for more than 9,600 genes (54), including deletion cassettes for genes in either of the two mating types, were acquired from the Fungal Genetic Stock Center (113). From the KO strains stock, the 126 available KO mutants for genes in the predicted SMCs (Table S1) were examined for altered phenotypes during asexual and sexual growth and development. All strains were cultured on BM for asexual growth and on SCM for sexual growth under constant white light at 25°C. Three independent investigations were applied to verify phenotypes observed for KO strains.

**Data availability.** Transcriptomics data generated from the Townsend lab and used in this study have been deposited in the GEO database under accession no. GSE41484, GSE101412, and GSE168995.

## SUPPLEMENTAL MATERIAL

Supplemental material is available online only.

**FIG S1**, TIF file, 0.2 MB.
**TABLE S1**, XLSX file, 0.2 MB.
**TABLE S2**, CSV file, 2.6 MB.
**TABLE S3**, XLSX file, 0.1 MB.
**TABLE S4**, XLSX file, 0.02 MB.
**TABLE S5**, XLSX file, 0.03 MB.

## ACKNOWLEDGMENTS

We thank the Broad Institute, JGI, and FungiDB for making *Neurospora crassa* genomic and phenotype data available and Rudy Diaz for a helpful edit of the manuscript.

We declare that no competing interests exist.

This study was supported by funding to J.P.T. from the National Institutes of Health R01 grant AI146584, the National Science Foundation grant IOS 1457044 to J.P.T., the National Science Foundation grant IOS 1456482 to F.T., and DEB-1638999 to J.S. O.Y., Z.W., and J.P.T. were supported by funding from the Binational Israel-U.S. Science Foundation grant number 2018712. The funders had no role in study design, data collection and interpretation, or the decision to submit the work for publication.

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
