## [Reviewer comments · mSystems]

Secondary metabolism gene clusters exhibit increasingly dynamic and differential expression during asexual growth, conidiation, and sexual development in *Neurospora crassa*

Zheng Wang, Francesc Lopez-Giraldez, Jason Slot, Oded Yarden, Frances Trail, and Jeffrey Townsend

Corresponding Author(s): Jeffrey Townsend, Yale School of Public Health

Review Timeline:

Submission Date:	March 7, 2022
Editorial Decision:	April 19, 2022
Revision Received:	April 25, 2022
Accepted:	April 29, 2022

Editor: Laura Hug

Reviewer(s): The reviewers have opted to remain anonymous.

Transaction Report:

DOI: <https://doi.org/10.1128/msystems.00232-22>

April 19, 2022

Prof. Jeffrey Peter Townsend
Yale School of Public Health
Biostatistics
135 College St, Suite 200, #222
New Haven, CT 06510

Re: mSystems00232-22 (Secondary metabolism gene clusters exhibit increasingly dynamic and differential expression during asexual growth, conidiation, and sexual development in *Neurospora crassa*)

Dear Prof. Jeffrey Peter Townsend:

Thank you for submitting your manuscript to mSystems. We have completed our review and I am pleased to inform you that, in principle, we expect to accept it for publication in mSystems. However, acceptance will not be final until you have adequately addressed the reviewer comments.

The reviewers have identified a few minor points for clarification. Please note Reviewer #2's suggestion to expand tables to summarize more information, as this would be extremely helpful for your readership.

Preparing Revision Guidelines

Sincerely,

Laura Hug

Editor, mSystems

Journals Department
Reviewer comments:

Reviewer #1 (Comments for the Author):

The authors examined the expression of genes in secondary metabolite clusters and phenotypes of gene deletion mutants. The manuscript is lengthy but is well-organized and easy to follow. My comments and questions are below.

L194. "Higher expression correlation of SMC ... with SMCs #3, #5, and #11 or SMCs #4, #10, #14, #15, and #18"
In Fig. 1 I, SMCs #3, #5, and #11 do not appear to cluster together.

L220 "Genes in SMCs #1, #11 and #15 exhibited similar patterns of expression, as did SMCs #2, #3, #4, #8, #9 and #17. The expression of genes in all other SMCs exhibited a third distinct pattern (Fig. 1K)"
Not sure if I understood the argument. In Fig. 1K, SMCs #7, #13, #18 and #20 (a subset of other SMCs) do form a distinct third pattern. However, many other SMCs, such as #3 and #9 are in the first cluster, and #19 and #5 are in the second cluster.

L306 "rhythmic with a period of 10-11 hours"
Doesn't conidiation occur diurnally (a period of 24 hours)? Please verify.

L380 "However, assessed knockouts of the pks-like genes"
I wonder whether neurosporin A was produced in the KO strains.

L448 "genes in the clusters Therefore"
A period is missing.wqq

Reviewer #2 (Comments for the Author):

This is a comprehensive analysis of the function of secondary metabolite genes and their role in asexual and sexual development in *Neurospora crassa*. The authors have performed an in depth genomic analysis of secondary metabolite gene clusters in *Neurospora crassa* and prepared and analyzed several experiments detailing gene expression. They also assess the phenotypes of secondary metabolite gene deletions from the *N. crassa* genome wide deletion project. Overall, this is a major contribution to the study of secondary metabolites in *Neurospora crassa*.

Concerns:

There is a considerable amount of data included in this manuscript. It would be useful to expand Table 2 to include a summary of the expression and other data. For example, which clusters have "orphan" genes? Which clusters were expanded by looking for co-expressing genes not identified by antiSMASH? For which clusters have there been described products w/ chemical structure? Is it possible to summarize expression patterns somehow in this table. In short, there is a lot of data and it gets cumbersome to keep track of all of the different expression patterns, etc, while reading the text.

Are all of the deletions with phenotypes noted in the text of the paper? A table of all genes whose deletion leads to a phenotype would be helpful.

We deeply appreciate the time and constructive efforts and suggestions both reviewers invested to improve our manuscript. We carefully addressed all the comments and concerns point-by-point as listed below and expanded table 2 as reviewer #2 suggested to include more detailed information to help readers better understand the described research.

Reviewer comments:

Reviewer #1 (Comments for the Author): The authors examined the expression of genes in secondary metabolite clusters and phenotypes of gene deletion mutants. The manuscript is lengthy but is well-organized and easy to follow. My comments and questions are below.

L194. "Higher expression correlation of SMC ... with SMCs #3, #5, and #11 or SMCs #4, #10, #14, #15, and #18" In Fig. 1 I, SMCs #3, #5, and #11 do not appear to cluster together.

>> Thank you for pointing out this discrepancy. Indeed, SMCs #3, #5, and #11 did not appear to cluster closely in PDA cultures, in contrast to BM and MSM cultures. We have revised the text so that it now accurately reads, "Higher expression correlation of SMCs was observed when *N. crassa* was cultured on BM and MSM at 25°C than when it was cultured on BM at 37°C, with SMCs #3, #5, and #11 or SMCs #4, #10, #14, #15, and #18 exhibiting two distinct expression patterns, and SMCs #4, #10, #14, #15, and #18 also exhibited a similar expression pattern on PDA at 37°C (Fig. 1G–I)."

L220 "Genes in SMCs #1, #11 and #15 exhibited similar patterns of expression, as did SMCs #2, #3, #4, #8, #9 and #17. The expression of genes in all other SMCs exhibited a third distinct pattern (Fig. 1K)" Not sure if I understood the argument. In Fig. 1K, SMCs #7, #13, #18 and #20 (a subset of other SMCs) do form a distinct third pattern. However, many other SMCs, such as #3 and #9 are in the first cluster, and #19 and #5 are in the second cluster.

>>The reviewer is correct: there were largely two patterns that can be recognized for SMCs during the conidiation process in Fig. 1K, which was one of the key points in our conclusions. Accordingly, we have revised the text as the reviewer suggested: "SMCs manifested two clustered patterns during conidiation. Genes in SMCs #1, #3, #9, #11 and #15 exhibited similar patterns of expression, and the expression of genes in all other SMCs exhibited another distinct pattern (Fig. 1K)."

L306 "rhythmic with a period of 10-11 hours" Doesn't conidiation occur diurnally (a period of 24 hours)? Please verify.

>> The diurnal rhythm of conidiation can be attributed to Gooch et al. 2004. The finding in Gooch et al. 2004 there is a rhythm to the higher vs lower rate of conidiation, rather than a discretely on-off rhythm of conidiation. To quote Gooch et al, 2004: "the conidiation rate would be high and constant for 10 to 11 h while a band was forming and would drop to a low constant rate while an interband was forming". So the rhythm of *N. crassa* between middle points of two interbands or two conidial bands is about 22 to 24 hours, but the rhythm between two conidial bands will be 10–11 hours. To avoid confusion

on this point, we revised our text so that it now reads “High-rate production of conidia that forms a conidiation band in *N. crassa* is rhythmic, with a period of 10–11 hours at 25°C. Thus, peak production rate of conidia occurs about 12 h after inoculation of mycelia (86)”.

L380 "However, assessed knockouts of the pks-like genes" I wonder whether neurosporin A was produced in the KO strains.

>>Good point! We are not able to assess biochemical elements for KO phenotypes in this study. However, we have revised the text to mention the opportunity: “However, assessed knockouts of the *pks*-like genes in SMC #3 did not manifest phenotypes in either asexual or sexual development. Therefore, a further biochemical assay would be warranted, investigating the effects of this gene knockout on the production of neurosporin A”

L448 "genes in the clusters Therefore" A period is missing.wqq

>> Thank you for noting this: the period has been added.

Reviewer #2 (Comments for the Author):

This is a comprehensive analysis of the function of secondary metabolite genes and their role in asexual and sexual development in *Neurospora crassa*. The authors have performed an in depth genomic analysis of secondary metabolite gene clusters in *Neurospora crassa* and prepared and analyzed several experiments detailing gene expression. They also assess the phenotypes of secondary metabolite gene deletions from the *N. crassa* genome wide deletion project. Overall, this is a major contribution to the study of secondary metabolites in *Neurospora crassa*.

Concerns: There is a considerable amount of data included in this manuscript. It would be useful to expand Table 2 to include a summary of the expression and other data. For example, which clusters have "orphan" genes? Which clusters were expanded by looking for co-expressing genes not identified by antiSMASH? For which clusters have there been described products w/ chemical structure? Is it possible to summarize expression patterns somehow in this table. In short, there is a lot of data and it gets cumbersome to keep track of all of the different expression patterns, etc, while reading the text.

>> We have followed the reviewer’s suggestion to expand table 2 with two more columns (columns 6 and 7) and one modified column (column 3) to summarize the data points in this study. The revised table 2 includes information regarding orphan genes and clusters with co-expressing neighbor genes that were not identified by antiSMASH, reporting two key pieces of information that are not directly reported in the main manuscript. Since *N. crassa* is not a traditional SMC model and biochemical assays for SMC genes are beyond the scope of this study, we reported no chemical structures for any SMC gene products—except for a few that have been characterized in previous studies cited in the text. On further

consideration of specific expression patterns that manifested across different life cycles under different conditions, we concluded that the provided figures of expression profiles better serve the informational purpose than wordy descriptions in a table would. However, to help the readers to track the results, we added environmental and developmental conditions where Coordinately Regulated Gene Sets (CRGS) were identified for each SMC in the revised Table 2 as described below. Lastly, new information in the revised Table 2 was carefully checked and cited properly in the revised text.

Table 2. Core genes of secondary metabolism clusters in *N. crassa*.

SMC#	Chromosome	antiSMASH clusters ^a	JGI ^b	SMC type ^c	CRGS ^d	Notes ^e
1	I (7914549–7932266)	NCU00583–00587 (NCU00583)	Not identified	PKS	BM, PDA, BM37	NCU00582, 00589 (BM)
2	I (8432960–8474209)	NCU03000–03016	NCU03010	NRPS-like	BM, SCM	None
3	I (88672729–8717074)	NCU16468, 02913–02926	NCU02927–02918	PKS	VM, SCM	NCU02908, 02910, 02930-31 (SCM)
4	II (3937420–3969336)	NCU08402–08403, 16586, 08404–08409, 16588, 08410	NCU08404–08407	NRPS-like	BM	NCU08411-13 (BM); 08411-12 (SCM)
5	II (4062839–4096852)	NCU08436–08443, 08445 (NCU08442)	NCU08439–08443	NRPS	No condition	None
6	IV (1967–36181)	NCU10285, 09635–09641, 10572, 09642	NCU10285, 09635–09640	PKS	BM, BM37, VM	NCU09627-30, 09633-34 (BM)
7	IV (508445–548448)	NCU04860–04862, 04865–04867 (NCU04867)	NCU04865	PKS	PDA	None
8	IV (4422480–4439201)	Not identified	NCU07307–07308	PKS-like	All conditions	NCU07310 (BM)

9	V (1702664–1723869)	NCU03583–03585	Not identified	PKS	PDA, BM37	NCU03582 (SCM)
10	V (4020488–4031764)	NCU01423–01427	Not identified	Terpene	PDA	None
11	VI (59272–103867)	NCU07119–07126, 07117 (NCU07122)	NCU07119	NRPS	VM	None
12	VI (398436–416452)	NCU04692, 17064, 12075, 04694–04695	NCU12075	DMAT	MSM, BM37	NCU04699 (BM); 04690, 04691, 04697 (SCM)
13	VI (854581–890466)	NCU04797–04806 (NCU04804)	Not identified	PKS-like	BM	NCU04796 (BM)
14	VI (3080467–3107588)	NCU06013, 17123, 06007–06012	NCU06013	PKS	BM37	NCU06001, 06005, 06019 (BM)
15	VI (4070819–4105738)	NCU05007–05014	NCU05011	PKS	BM, PDA, BM37	NCU05006, 05015 (SCM)
16	VI (4116143–4157329)	NCU05005–05004, 10597, 05001, 12150–12152, 05000, 12154, 04998–04996, 10537, 04994, 12156 (NCU04998)	NCU05000, 12154	NRPS-like	No condition	NCU04992 (BM); 04991-92 (SCM)
17	VII (1083585–1110519)	NCU08395–08399, 12034	NCU08399	PKS	BM37	NCU08390 (BM)
18	VII (1529996–1574489)	NCU04528–04535, 12022–12021, 06170	NCU04531	NRPS	BM37, SCM	NCU04520-24, 04537 (BM)
19	VII (1954064–1973561)	NCU06054–06056, 10683, 06052–06051	Not identified	Terpene	BM, VM	NCU06057 (BM)

20	VII (3720911–3763243)	NCU05755–05763, 17267, 05764, 17268–17271, 05766–05769 (NCU05759)	Not identified	RiPP	MSM, VM, SCM	NCU05770 (BM); 05751-52, 05773-74 (SCM)
----	--------------------------	--	----------------	------	--------------	--

^a Orphan genes are listed in parentheses with a bolded font.

^b SMCs reported at the JGI Mycocosm database (Grigoriev *et al.*, 2014).

^c SMC-type identification follows the glossary in the antiSMASH documentation.

^d Environmental and developmental conditions in which Coordinately Regulated Gene Sets (CRGS) were identified, including asexual growth on BM, MSM, and PDA at 25°C, asexual growth on BM at 37°C (BM37), conidiation on Vogel’s medium (VM), and sexual development on SCM at 25°C (as in **Fig. S1**).

^e An additional five genes on each side of the predicted SCM were examined to ascertain whether they exhibited the same expression patterns as any two or more of the SCM core genes during asexual growth on BM at 25°C and/or during sexual development on SCM at 25°C.

Are all of the deletions with phenotypes noted in the text of the paper? A table of all genes whose deletion leads to a phenotype would be helpful.

>>We thank the reviewer for this thoughtful suggestion. Phenotypes for knockout strains available for 126 SMC genes have now been added to the revised Table S1, including observations based on asexual growth on BM and sexual development on SCM. Since knockout phenotypes observed for several SMC genes in this study were consistent with previous studies (cited in the text), they were addressed concisely in the text in the results and discussion section under subtitle “Some SMC genes exhibited phenotypes when knocked out”.

April 29, 2022

Prof. Jeffrey Peter Townsend
Yale School of Public Health
Biostatistics
135 College St, Suite 200, #222
New Haven, CT 06510

Re: mSystems00232-22R1 (Secondary metabolism gene clusters exhibit increasingly dynamic and differential expression during asexual growth, conidiation, and sexual development in *Neurospora crassa*)

Dear Prof. Jeffrey Peter Townsend:

Thank you for returning your revised manuscript, and for your thorough attention to detail in addressing the reviewers' comments.

Your manuscript has been accepted, and I am forwarding it to the ASM Journals Department for publication. For your reference, ASM Journals' address is given below. Before it can be scheduled for publication, your manuscript will be checked by the mSystems production staff to make sure that all elements meet the technical requirements for publication. They will contact you if anything needs to be revised before copyediting and production can begin. Otherwise, you will be notified when your proofs are ready to be viewed.

Publication Fees:

We recognize that the video files can become quite large, and so to avoid quality loss ASM suggests sending the video file via <https://www.wetransfer.com/>. When you have a final version of the video and the still ready to share, please send it to mSystems staff at mSystems@asmusa.org.

For mSystems research articles, if you would like to submit an image for consideration as the Featured Image for an issue, please contact mSystems staff at mSystems@asmusa.org.

Sincerely,

Laura Hug
Editor, mSystems

Journals Department
Fig. S1: Accept
Table S2: Accept
Table S1: Accept
Table S3: Accept
Table S4: Accept
Table S5: Accept